# A Simplified Low-Dose 10-Day Quadruple Therapy with a Galenic Formulation of Bismuth Salicylate Is Highly Effective for *Helicobacter pylori* Eradication

**DOI:** 10.3390/jcm12020681

**Published:** 2023-01-15

**Authors:** Maria Pina Dore, Francesco Saba, Lucia Zanni, Anna Rocca, Jessica Piroddu, Giuseppe Gutierrez, Giovanni Mario Pes

**Affiliations:** 1Dipartimento di Medicina, Chirurgia e Farmacia, Clinica Medica, University of Sassari, Viale San Pietro 8, 07100 Sassari, Italy; 2Department of Medicine, Baylor College of Medicine, One Baylor Plaza Blvd, Houston, TX 77030, USA; 3Pharmaceutical Laboratory “Gutierrez-Pisano”, 07100 Sassari, Italy

**Keywords:** *Helicobacter pylori* infection, quadruple therapy, bismuth salicylate

## Abstract

Background: Earlier studies have shown that a modified low-dose bismuth quadruple therapy given for 10 to 14 days is highly effective for the treatment of *Helicobacter pylori* infection in Sardinia. However, bismuth is not universally available. Aim: We aimed to investigate the efficacy of a simplified low-dose 10-day quadruple therapy containing a galenic formulation of bismuth salicylate for *H. pylori* infection. Patients and Methods: Adult patients positive for *H. pylori* infection were assigned to a quadruple therapy containing a galenic formulation of bismuth salicylate (200 mg) plus tetracycline 500 mg, metronidazole 500 mg and rabeprazole 20 mg, given twice a day with the midday and evening meals for 10 days. A negative stool antigen test or 13C-Urea Breath Test defined successful eradication. Compliance and adverse events were recorded 30–40 days after the end of treatment. Results: In this open-label pilot study, 42 patients were enrolled (mean age 54.1 ± 12.0 years; 64% female). Among the study participants, 35 were naïve to *H. pylori* treatment. The treatment regimen was completed by 41 patients, with an overall success rate of 95.1%. More specifically, the eradication rate was 95.1% PP; 95% confidence interval (CI) = 86.6–100 and 92.9% by ITT; 95%CI = 85.1–100%, respectively. For naïve patients, the cure rate was 97.1%. Compliance was excellent. Side effects were absent or mild overall. Conclusions: The modified low-dose 10-day quadruple therapy provided high eradication rates of *H. pylori* infection, despite the replacement of colloidal bismuth subcitrate with bismuth salicylate. In regions where bismuth is unavailable in the market, the galenic formulation should be a valid option.

## 1. Introduction

*Helicobacter pylori* (*H. pylori*) infection is one of the most common chronic bacterial infections affecting mankind [1]. In the last few decades, several antibiotic regimens have been proposed and evaluated in different populations to treat the infection, although few regimens have consistently achieved satisfying eradication rates, especially because of increasing antibiotic-resistance rates [2,3,4,5]. For example, triple therapies based on clarithromycin, metronidazole or levofloxacin should be used only on the basis of susceptibility [2,3,4,5]. Similarly, non-bismuth quadruple therapies consisting of a combination of proton pump inhibitors (PPIs), amoxicillin, metronidazole/tinidazole and clarithromycin, taken together or sequentially (known as concomitant therapy, sequential therapy and hybrid sequential–concomitant therapy), are currently considered obsolete [2,3,4,5]. These regimens are not recommended anymore because at least one antibiotic for each patient is unnecessary [6,7,8].

Ideally, similarly to other bacterial infections, the choice of antibiotic regimen to treat *H. pylori* for a naïve patient should be guided by susceptibility tests, nowadays available for clarithromycin, levofloxacin, metronidazole, amoxicillin, tetracycline and rifabutin, through next-generation sequencing on gastric biopsies (fresh or formalin-fixed), stool samples and culture [9]. However, this approach is still a mirage in several countries, continuing the usual management to treat the infection empirically. According to the last Maastricht consensus, when individual susceptibility testing is not available, the first-line recommended treatment should be bismuth quadruple therapy (BQT), in areas of high clarithromycin resistance [3]. The BQT therapy was originally a combination of a PPI, a bismuth, tetracycline HCl and metronidazole given for 14 days [10,11]. At that time, authors used bismuth subcitrate 120 mg plus tetracycline 500 mg both q.i.d. for 28 days and metronidazole 200 mg q.i.d. for 14 days, with a cure rate of 94% [10,11]. Nowadays, cure rates greater than 90 % to 95 % are expected with BQT regimens, although nearly 50% of patients will experience side effects, such as diarrhea, nausea, bad taste, abdominal pain, etc., inducing some of them stopping the treatment [12]. Despite metronidazole resistance, this regimen proved to be effective, because resistance can often be overcome by using a higher dose of metronidazole therapy or a longer treatment duration. However, this regimen is troublesome because of the need to take multiple pills several times a day. Bismuth quadruple therapy is also available combined in a capsule (Pylera), without the PPI. The three-in-one capsule (three capsules four times daily plus a PPI twice daily) could have the advantage of increasing patient adherence compared to traditional standard quadruple therapy (four to eight pills q.i.d. and a PPI b.i.d.).

In earlier studies conducted in Sardinia, a region characterized by a high prevalence of clarithromycin, metronidazole and amoxicillin resistance [13,14], we observed that a modified low-dose colloidal bismuth subcitrate-based quadruple therapy, given for 14 days with the evening and midday meals, to enhance compliance, was able to achieve *H. pylori* eradication of 95% (95% CI = 90–98%) by the intention to treat (ITT) and 98% per protocol (PP) analysis, irrespective of prior treatment failure [15]. The same regimen showed excellent cure rates despite a shorter duration (from 14 to 10 days) in a head-to-head trial [16].

Recently, Geng et al. reported a successful suppression with bismuth subsalicylate given in monotherapy of a drug-resistant *H. pylori* strain [17].

Because the colloidal bismuth subcitrate (De-Nol) is no longer available in Italy as an individual therapy, and the three-in-one capsule Pylera (Pylera^®^; Allergan, Dublin, Ireland) is difficult for patients to obtain, the purpose of this study was to evaluate the efficacy of a simplified low-dose BQT containing a galenic formulation of bismuth salicylate to treat *H. pylori* infection.

## 2. Materials and Methods

### 2.1. Study Design

This was a prospective, single-center, open-label trial conducted in adult patients from Northern Sardinia positive for *H. pylori* infection, to evaluate the efficacy and tolerability of BQT containing a galenic formulation of bismuth salicylate. The study was conducted at the department of internal medicine, gastroenterology section, University of Sassari, Sassari, Italy.

### 2.2. Participant’s Eligibility

Patients attending a gastroenterology visit and found to be positive for *H. pylori* infection by histology collected during an upper endoscopy or by ¹³C-UBT or the antigen stool test were invited to participate in the study. At baseline, for each patient, data, including age, sex, cigarette smoking and body height and weight, were collected. Complained symptoms, history of allergy to any of the drugs used and prior treatment for *H. pylori* infection were also recorded.

Considerable time was spent for each visit to explain the importance and benefits of therapy and the expected side effects according to the education level, age and gender of the individual patient. Moreover, a clear written and verbal description (sometimes in the local dialect) of the medications and plan for dosing were provided [18].

Patients younger than 18 years or with a history of drug or alcohol abuse were excluded. The presence of clinically significant comorbidities or malignancy, pregnancy or lactation as well as known allergy to any component of the regimen used in the study were considered exclusion criteria. The treatment was delayed for at least one month in patients who took antibiotics or probiotics during the four weeks preceding the enrollment. Patients returned 4 to 6 weeks after the end of treatment for the first follow-up visit to check *H. pylori* eradication, to assess compliance and side-effect occurrence.

### 2.3. H. pylori Status

Pre-treatment *H. pylori* infection was defined as the presence of *H. pylori* on histological examination of gastric biopsies (2 from the antrum, 1 from the angulus and 1 from the gastric corpus) or a positive ¹³C-UBT or antigen stool test. Post-treatment success was defined by a negative ¹³C-UBT or a negative *H. pylori* antigen stool assay at least 30 days after therapy. Pre-treatment culture of biopsy specimens and antibiotic susceptibility tests were not performed.

### 2.4. Treatment Regimen

The modified low-dose 10-day BQT consists of rabeprazole 20 mg, metronidazole 500 mg, tetracycline hydrochloride 500 mg plus bismuth salicylate 200 mg, all given b.i.d. with the midday and evening meals for a total of 10 days (Table 1).

The galenic formulation of bismuth salicylate was prepared in a local pharmaceutical laboratory (Gutierrez-Pisano Pharmacy, 07100 Sassari, Italy). Briefly, the bismuth salicylate and excipients, including pregelatinized starch, anhydrous colloidal silica and magnesium stearate (Farmalabor Company, Canosa di Puglia, 76012 BT, Italy), were carefully weighed and reduced to 300 micron granules to mix trough geometric dilutions in a mortar. An additional step was carried out in a rotary mixer until homogeneity. After checking mass uniformity, capsules were filled, packaged and labelled.

### 2.5. Ethical Considerations

Verbal and written informed consent were obtained from each participant and the study was approved by the local Institutional Review Board “Comitato Etico ASL n. 1 di Sassari” (Prot. 2358/CE). The study was performed according to good clinical practice and the Declaration of Helsinki.

### 2.6. Statistical Analysis

The eradication rate of *H. pylori* infection was assessed via ITT analysis, including all eligible patients enrolled in the study, and by PP analysis, which excluded patients lost to follow-up. Adherence to the treatment was labelled as excellent when treatment was taken for at least nine days. Ninety-five percent confidence intervals (95%CIs) were also calculated, and the significance threshold was set at *p* < 0.05.

## 3. Results

In total, 42 consecutive patients were included in the study. All participants were from Northern Sardinia, Italy. Thirty-five subjects (mean age 54.1 ± 12.0 years; 27 female) were naïve for *H. pylori* treatment. Among patients, 8 were current and 13 former smokers. Interestingly, gastric erosions were observed in only one patient at the upper endoscopy. Intestinal metaplasia was present in gastric specimens from the antrum of two patients among the 21 undergoing endoscopy (Table 2). Seven participants were previously treated for *H. pylori* by traditional PPI–amoxicillin and clarithromycin regimens.

The treatment was completed by 41 patients (1 patient discontinued the treatment because of the onset of an allergy) and the simplified low-dose BQT was effective in 95% (39/41) of patients (Table 3).

More specifically, the eradication rate was 95.1% PP (95.1%CI = 88.6–100%) and 92.9% for ITT (95%CI = 85.1–100%), respectively. Correlations between BMI, smoking habits, age, sex, previous failure treatment and cure rate were not detected. Treatment-failure participants were treated with BQT at the traditional full dosage, with a cure rate of 100% (two/two patients) (Table 3).

Moreover, the simplified low-dose BQT regimen showed an excellent performance in naïve patients as a first-line treatment (97.1%). As a rescue therapy, it was less effective (83.3%). However, the small number of failure patients did not allow for any definite conclusion (Table 4).

The overall tolerability was good and the reported compliance was excellent for each participant. Side effects were, overall, mild and included diarrhea, bad taste and abdominal discomfort.

## 4. Discussion

Curing *H. pylori* infection results in gastritis healing, elimination of peptic ulcer risk and, based on the reversibility of the gastric mucosa damage that has occurred, a reduction in the risk of developing gastric cancer [19]. In the last few decades, several antibiotic regimens have been proposed and evaluated in different populations, clinical trials and meta-analyses to treat *H. pylori* infection [20,21]; however, few regimens have consistently achieved satisfying eradication rates, especially because of antibiotic-resistance rates [9,21].

The correct choice of a regimen requires a minimum knowledge of the evidence in this field, a detailed clinical history of the individual patient about previous exposure to antibiotics and for specific allergies and local antibiotic-resistance patterns. Adverse events, dosage, administration modality and availability are also important issues to take into account when choosing a regimen [9].

Similar to other infections, to achieve high cure rates, patient-specific susceptibility testing is needed. If there are no options other than to give empiric therapy, all consensus uniformly recommends using only those regimens proven to be effective locally [3,9].

Because of increasing resistance, triple therapies based on clarithromycin, metronidazole or levofloxacin should be used on the basis of susceptibility. In most regions, 14-day four-drug combinations are required for successful empiric therapy [2,3,9].

Bismuth quadruple therapy includes bismuth subcitrate or subsalicylate, metronidazole, tetracycline HCl plus a PPI. The eradication rate increases with administration for 14 days in association with a second-generation PPI (esomeprazole and rabeprazole) given in a double dose (equivalent to at least 40 mg of omeprazole and minimally affected by CYP2C19 metabolism) [22]. The three-in-one capsule Pylera four times daily plus a PPI twice daily, given for 10 days, was able to achieve a mean eradication rate of 91% in the US and Europe [23,24].

In previous studies conducted in Sardinia, legacy clarithromycin or amoxicillin containing triple therapy provided discouraging eradication rates, justified by the high prevalence of pretreatment antibiotic resistance [13,14]. However, in the same region, metronidazole containing triple therapy showed a high success rate, despite metronidazole-resistant *H. pylori* strains [13]. The addition of bismuth (colloidal bismuth subcitrate) to that regimen was able to notably increase the eradication rate [15]. Exploratory studies confirmed that the efficacy of BQT was maintained, despite a reduced dosage and treatment duration (metronidazole 500 mg and tetracycline 500 mg twice a day for 10 days). The simplified low-dose regimen proved to be highly effective in adult and elderly infected patients as a first-line and salvage therapy, with the advantage of fewer side effects and a lower amount of antibiotics [16]. In addition, patient adherence to the treatment was enhanced by the fact that medicines were taken with food (during meals).

However, in 2016, De-Nol was removed from the Italian market and the Pylera^®^ patent has expired, making it difficult for the doctor to depict the best regimen. In order to avoid patient exposure to treatment regimens already known for their poor effectiveness against *H. pylori*, the modified low-dose regimen was supplemented with a galenic formulation of bismuth salicylate, to take advantage of the topical action against the bacteria in the gastric mucosa. In fact, bismuth salts have a bactericidal effect on *H. pylori* and bismuth-containing therapy was demonstrated to be an effective treatment option in populations with multi-resistant *H. pylori* strains [3].

Furthermore, salicylate is able to reversibly block the synthesis of flagellin and flagella of enteric Gram-negative bacteria [25]. There is evidence in the literature that salicylate is also able to suppress the motility of *H. pylori* flagella [26] and, in turn, the colonization of the gastric mucosa. Accordingly, mice immunized with a vaccine against flagella of *H. pylori* significantly reduced the bacteria colonization [27].

## 5. Conclusions

Our results showed that simplified low-dose BQT containing a galenic formulation of bismuth salicylate is equally effective to the simplified low-dose BQT containing colloidal bismuth subcitrate. Replacement of colloidal bismuth subcitrate with a homemade bismuth salicylate formulation should be a good option for regimens against *H. pylori* when bismuth is unavailable on the market.

## Figures and Tables

**Table 1 jcm-12-00681-t001:** Dosage and schedule of treatment given for 10 days.

B.I.D. Bismuth	Lunch	Dinner
Metronidazole 250 mg	2 caps	2 caps
Tetracycline 250 mg	2 caps	2 caps
Rabeprazole 20 mg	1 cap	1 cap
^1^ Bismuth	1 cap	1 cap

^1^ The galenic bismuth formulation: 1 cap contains 200 mg of bismuth salicylate.

**Table 2 jcm-12-00681-t002:** Baseline characteristics of treated patients.

Characteristics	
No. of patients	42
Male/female	15/27
Mean age (yrs)	54
Naïve	35
Smokers	19%
Former smokers	30%
Body mass index (range)	17–40 kg/m²
Peptic ulcer/erosions	1
Intestinal metaplasia	2/21

**Table 3 jcm-12-00681-t003:** Intervention status of patients enrolled in the study.

Status	Patients
Received intervention	42
Drop-out	1
Completed trial	41
Intervention ineffective	2
Cure rate ITT ^1^	92.9% (39/42)
95% CI ^#^	85.1–100
Cure rate PP ^2^	95.1% (39/41)
95% CI ^#^	88.6–100

^1^ ITT: intention-to-treat analysis; ^2^ PP: per protocol analysis; ^#^ CI: Confidence Interval.

**Table 4 jcm-12-00681-t004:** Prevalence of cure rates per protocol according to failure or naïve patients for *H. pylori* treatment.

Naïve Patients (35)	97.1% (34/35)
95% CI ^#^	(91.6–100)
Failure patients	83.3% (5/6)
95% CI ^#^	(53.5–100)

^#^ CI: Confidence Interval.

## Data Availability

Data supporting reported results can be available upon request.

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
