# Peer review of "A Simplified Low-Dose 10-Day Quadruple Therapy with a Galenic Formulation of Bismuth Salicylate Is Highly Effective for Helicobacter pylori Eradication"

_jcm, 2023, doi:10.3390/jcm12020681_

Round 1

Reviewer 1 Report

The manuscript presents an open study of the effect of a galenic bismuth formula on H. pylori eradication. My concerns are:

1. trivial considerations on the role of H.,pylori from the introduction could be deletes: everybody knows that it causes ulcer etc.....

2. the statement that bismuth is unavailable in all Europe is erronneous

3. tinidazole is also unavailable in many countries, although it has more advantageous pharmacologic profile than metronidazole. Could you detail the reasons why these drugs have limited accesibility?

4. The number of patients include is rather low, why? Sardinia has a rather high H. pylori prevalence! were there financial constraints?

5. The eradication results were not calculated on an modified ITT basis

6. References: item 6: in the meantime, the Maastricht VI/Florence consesus was published and might be replaced

7. The proportion of self-citation, aslthoughbnappropriate,  is 27% (8 titles from 30), it seems rather high.

Author Response

Reviewer # 1

The manuscript presents an open study of the effect of a galenic bismuth formula on H. pylori eradication. My concerns are:

trivial considerations on the role of H. pylori from the introduction could be deletes: everybody knows that it causes ulcer etc

Reply. We agree with the reviewer that some considerations about H. pylori are trivial, however they were added to the introduction just to be consistent with the majority of papers in this field. Anyway it was a pleasure to remove them.

The statement that bismuth is unavailable in all Europe is erroneous.

Reply. It is correct, thank you. The sentence was changed accordingly whenever necessary across the manuscript.

tinidazole is also unavailable in many countries, although it has more advantageous pharmacologic profile than metronidazole.

Reply. This is true, especially for bacterial vaginosis. However, there are no head-to-head studies (tinidazole vs metronidazole) for H. pylori eradication in order to quantify the efficacy or fewer side effect of tinidazole vs metronidazole.

Could you detail the reasons why these drugs have limited accesibility?

Reply. We guess for marketing reasons, maybe because, as the colloidal bismuth, is too cheap, meaning little or no profit for the Pharma. However, we found to detail this issue in our paper  will be out of the topic, considering that tinidazole was not used in our regimen

The number of patients include is rather low, why? Sardinia has a rather high H. pylori prevalence!

Reply. Correct. In Sardinia the prevalence of H. pylori infection was very high at the end of the 1990s (62% among blood donors), in 1993 was 64% in dyspeptic patients and decreased to 19% nowadays, especially in the elderly. The median age of our studied cohort was 54 years, for this reason, the studied cohort is small, although still representative of a pilot study.

were there financial constraints?

Reply. Not at all! As usual in the clinical routine, patients paid by their pocket for the medicines. As specified in the manuscript, there was no financial support for this study.

The eradication results were not calculated on an modified ITT basis.

Reply. We preferred to use the ITT analysis because the application of a modified ITT analysis is criticized by several authors “modified ITT analysis allows a subjective approach in entry criteria, which is highly hazardous to manipulation and consequently to bias” (Modified Intention to Treat: frequency, definition and implication for clinical trials, cochrane.org/2007; Modified intention to treat reporting in randomised controlled trials: systematic review BMJ 2010; 340)

References: item 6: in the meantime, the Maastricht VI/Florence consesus was published and might be replaced.

Reply. Thank you so much to notice it. The reference was replaced in the revised manuscript

The proportion of self-citation, aslthoughbnappropriate, is 27% (8 titles from 30), it seems rather high.

Reply. In order to build the history to make the rationale for the aim of our study it was necessary to cite our previous work in the field. However, in the revised manuscript we cut self-citation as much as we can (# 2, 26 and 27)

Reviewer 2 Report

The authors investigated the efficacy of a simplified low-dose 10-day quadruple therapy containing a galenic formulation of bismuth salicylate for H. pylori infection. They disclosed that the eradication rate was 95.1% PP and 92.9% ITT with excellent compliance and few to mild side effects. They concluded that, in regions where bismuth is unavailable in the market, the galenic formulation should be a valid option of H. pylori eradication. The novel regimen using a simplified low-dose 10-day quadruple therapy with a galenic formulation of bismuth salicylate for H. pylori infection the authors investigated in this manuscript was effective and the authors presented them well. This manuscript gives us instructive message. However, there were inconsistencies in data they showed in abstract and results part, i.e. 95.1% PP; 95%CI = 86.6-100 and 92.9-100 for ITT; 95%CI = 85.1-100%. Also, they did not showed cure rate of naïve patients in results part. These discrepancies should be corrected before resubmission.

Author Response

Reviewer # 2

The authors investigated the efficacy of a simplified low-dose 10-day quadruple therapy containing a galenic formulation of bismuth salicylate for H. pylori infection. They disclosed that the eradication rate was 95.1% PP and 92.9% ITT with excellent compliance and few to mild side effects. They concluded that, in regions where bismuth is unavailable in the market, the galenic formulation should be a valid option of H. pylori eradication. The novel regimen using a simplified low-dose 10-day quadruple therapy with a galenic formulation of bismuth salicylate for H. pylori infection the authors investigated in this manuscript was effective and the authors presented them well. This manuscript gives us instructive message.

Reply. Thank you very much for the appreciation of our manuscript

However, there were inconsistencies in data they showed in abstract and results part, i.e. 95.1% PP; 95%CI = 86.6-100 and 92.9-100 for ITT; 95%CI = 85.1-100%. Also, they did not showed cure rate of naïve patients in results part. These discrepancies should be corrected before resubmission.

Reply. Thank you so much to highlight the stupid mistake, in the revised manuscript numbers were fixed.

For naïve patients, a new table (Table 4), was added. Page 5 lines 173-184